# Calculation of Crystal-Solution Dissociation Constants

**DOI:** 10.3390/biom12020147

**Published:** 2022-01-18

**Authors:** Sergiy O. Garbuzynskiy, Alexei V. Finkelstein

**Affiliations:** 1Institute of Protein Research, Russian Academy of Sciences, 142290 Pushchino, Russia; sergey@phys.protres.ru; 2Biology Department, Lomonosov Moscow State University, 119192 Moscow, Russia; 3Biotechnology Department, Lomonosov Moscow State University, 142290 Pushchino, Russia

**Keywords:** dissociation constant, binding entropy, molecular crystals, amplitude of movements in crystals, Henry’s law constant, computational chemistry and biochemistry

## Abstract

The calculation of dissociation constants is an important problem in molecular biophysics. For such a calculation, it is important to correctly calculate both terms of the binding free energy; that is, the enthalpy and entropy of binding. Both these terms can be computed using molecular dynamics simulations, but this approach is very computationally expensive, and entropy calculations are especially slow. We develop an alternative very fast method of calculating the binding entropy and dissociation constants. The main part of our approach is based on the evaluation of movement ranges of molecules in the bound state. Then, the range of molecular movements in the bound state (here, in molecular crystals) is used for the calculation of the binding entropies and, then (using, in addition, the experimentally measured sublimation enthalpies), the crystal-to-vapor dissociation constants. Previously, we considered the process of the reversible sublimation of small organic molecules from crystals to vapor. In this work, we extend our approach by considering the dissolution of molecules, in addition to their sublimation. Similar to the sublimation case, our method shows a good correlation with experimentally measured dissociation constants at the dissolution of crystals.

## 1. Introduction

In spite of recent large successes in the prediction of native spatial structures of proteins [1,2], the prediction of the interactions of biomolecules, as well as their binding affinities, still remains a challenge [3]. Such predictions have both a fundamental and applied value—for instance, for computational drug design. One of the main parameters here is the dissociation constant of molecules, by definition related to their binding free energy. Some researchers focus on the enthalpy component of the free energy of binding [4,5,6,7], while others suggest that the main obstacle to a satisfactory estimate of the free energy of binding is the difficulty of taking into account its entropy component [8,9].

Both terms that compose binding free energy (enthalpy and entropy) can, in principle, be estimated by molecular dynamics methods [10,11]. Enthalpy (or energy) can be estimated using various force fields (see, for example, [12] and references therein). The entropy of binding can be estimated [13] by tracing a very long (until reaching a thermodynamic equilibrium) molecular dynamic trajectory of the motion of all atoms in a complex of bound molecules (for example, in a protein–ligand complex), and then in these molecules taken separately.

Moreover, in models that consider the aqueous solvent in an “explicit” (atomic) form, this “molecular dynamic” entropy takes into account both the configurational entropy of the molecules forming the complex and the entropy of the surrounding solvent [12], while in models where the solvent is represented in an “implicit” form (as a medium), only the configurational entropy of the molecules forming the complex is taken into account, and the entropy of the solvent is included into the potentials of interatomic interactions, in particular, hydrophobic and electrostatic (see [14,15,16] and references therein).

The configurational entropy present in both types of models can also be estimated by other, approximate ways: from a range of side group conformations (observed by X-ray and NMR in proteins [17,18,19] or obtained by optimizing these conformations using various force fields [20]), from an estimate of the surface of molecules hidden from water [21], and from the calculation of the modes of only elastic [22] or all [23] vibrations of the protein molecule and its ligand.

The main problem in molecular dynamics methods (along with the inaccuracy of the force fields used in the calculations) is the huge computational time of the calculation. In this case, the enthalpy of an individual state (molecule or complex) can be calculated rather quickly (using one or another force field), but to calculate the entropy, it is necessary to simulate the entire ensemble of configurations or at least a significant part of it, up to reaching a full thermodynamic equilibrium (moreover, for the complex and for each molecule separately), which, naturally, requires a very large computational time. The problem is further complicated by the fact that both the ligand and its binding site can be significantly deformed during the interaction [8,13,24].

We developed [25,26,27] an alternative very fast method of calculating the binding entropy and dissociation constants. Our approach is based on the evaluation of the movement range of molecules in the bound state. This calculated value (average range of movements of molecules in the bound state—we consider molecular crystals as a simple test case now) is used for the calculation of the binding entropy and dissociation constants (both to vapor and to solution).

In previous works [25,26,27], we considered a simple case of the equilibrium between molecular crystals and vapor (that is, a reversible sublimation). The comparison of the calculated binding entropies and crystal-to-vapor dissociation constants with the corresponding experimental data has shown [26] a good coincidence with the experiments (correlation coefficients exceeded 90% both for the binding entropy and for dissociation constants).

Here, we extend our approach to a consideration of the process of dissolving molecules into a solution, which is a more biologically interesting process than sublimation. Similar to the sublimation case, we show a good coincidence of the calculated dissociation constants with the experimental values for the solution.

## 2. Methods

### 2.1. Model and Approach

The dissociation constant for a complex of two particles is [28] $K_{D}=\frac{\left[C_{A}\right]\left[C_{B}\right]}{\left[C_{\mathrm{AB}}\right]}$, where $\left[C_{A}\right]$ and $\left[C_{B}\right]$ are the equilibrium concentrations of separate particles A and B, respectively, and $\left[C_{\mathrm{AB}}\right]$ is the equilibrium concentration of their complex. At $\left[C_{\mathrm{AB}}\right]=\left[C_{B}\right]$, $K_{D}=\left[C_{A}\right]$. For a reaction of the “crystal of N molecules ↔ molecule + crystal of N-1 molecules” type, $\left[C_{\mathrm{crystal}\;\mathrm{of}\;N\;\mathrm{molecules}}\right]=\left[C_{\mathrm{crystal}\;\mathrm{of}\;N-1\;\mathrm{molecules}}\right]$ in equilibrium, so that the concentration of molecules, $\left[C_{A}\right]$, in saturated vapor or in a saturated solution is equal to the corresponding dissociation constant.

In equilibrium between crystal and solution, the chemical potential of a mole of crystal-forming molecules is the same for these two states ($\mu_{\mathrm{crystal}}^{}=\mu_{\mathrm{solution}}^{}$). It has the sense to divide $\mu_{\mathrm{solution}}^{}$ into two parts:(1)$$\mu_{\mathrm{solution}}^{}\equiv \mu_{\mathrm{solution}}^{-}-T\cdot R\ln\left[V_{\mathrm{solution}}^{}\right],$$
where $\left[V_{\mathrm{solution}}^{}\left(\frac{L}{\mathrm{mol}}\right)\right]\equiv 1/\left[C_{\mathrm{solution}}\left(\frac{\mathrm{mol}}{L}\right)\right]$ is the volume per mole of crystal-forming molecules in solution, and $R\ln\left[V_{\mathrm{solution}}^{}\right]$ is the translational entropy per mole of these molecules in solution, while $\mu_{\mathrm{solution}}^{-}$ does not depend on the entropy of translational movements. Thus:(2)$$RT\ln\left[C_{\mathrm{solution}}\right]=\mu_{\mathrm{crystal}}^{}-\mu_{\mathrm{solution}}^{-}.$$

On the other hand, $\mu_{\mathrm{crystal}}^{}-\mu_{\mathrm{solution}}^{-}$ can be presented in the form:(3)$$\mu_{\mathrm{crystal}}^{}-\mu_{\mathrm{solution}}^{-}=(\mu_{\mathrm{crystal}}^{}-\mu_{\mathrm{vapor}}^{-})+\left(\mu_{\mathrm{vapor}}^{-}-\mu_{\mathrm{solution}}^{-}\right),$$
where $\mu_{\mathrm{vapor}}^{-}$ is the chemical potential of the molecule in vapor without the entropy of translational movements.

The same approach applied to the equilibrium between a crystal and vapor gave:(4)$$RT\ln\left[C_{\mathrm{vapor}}\right]=\mu_{\mathrm{crystal}}^{}-\mu_{\mathrm{vapor}}^{-},$$
so that:(5)$$\mu_{\mathrm{vapor}}^{-}-\mu_{\mathrm{solution}}^{-}=RT\ln\left(\frac{\left[C_{\mathrm{solution}}\right]}{\left[C_{\mathrm{vapor}}\right]}\right),$$
where $\frac{\left[C_{\mathrm{solution}}\right]}{\left[C_{\mathrm{vapor}}\right]}$ is the Henry’s law constant $k_{H,\mathrm{cc}}$ [29] when both concentrations $\left[C_{\mathrm{solution}}\right]$ and $\left[C_{\mathrm{vapor}}\right]$ are low (less than 1 mol/L). Thus, $\mu_{\mathrm{crystal}}^{}-\mu_{\mathrm{solution}}^{-}=RT\ln\left[C_{\mathrm{vapor}}\right]+RT\ln\left(k_{H,\mathrm{cc}}\right)$, and:
(6)$$\left[C_{\mathrm{solution}}\right]=\left[C_{\mathrm{vapor}}\right]\times k_{H,\mathrm{cc}},$$
that is:(7)$$K_{D,\mathrm{solution}}=K_{D,\mathrm{vapor}}\times k_{H,\mathrm{cc}}.$$

That is, to calculate the crystal-to-solution dissociation constant, we can calculate the equilibrium concentration of molecules in vapor and multiply it by Henry’s law constant, which one can take from published experimental data. The way to estimate $\left[C_{\mathrm{vapor}}\right]$ developed in our work [26] used the calculation of entropy [25] (see Figure 1) and an experimental estimate of enthalpy.

### 2.2. Entropy and Dissociation Constant Calculation

To evaluate sublimation entropy, we considered movements available in vapor but hindered in crystal. We considered four types of such movements (see lines (a)–(d) in Figure 1): (a) translational movements of the molecule as a whole; (b) rotational movements of the molecule as a whole; (c) rotations around a covalent bond with a very low or no potential barrier; (d) vibrations around a covalent bond with a moderately high potential barrier.

In a classic approximation, a decrease in entropy of 1 mol of vapor molecules at their fixation in a crystal can be calculated ([25,26], see also Appendix A) as follows:(8)$$-\Delta S_{\mathrm{subl}}\equiv S_{\mathrm{crystal}}-S_{\mathrm{vapor}}=R\ln\left[\frac{\delta x_{1}\delta x_{2}\delta x_{3}}{{\hat{V}}_{\mathrm{vapor}}\cdot e}\right]+R\ln\left[\frac{\delta \beta_{1}\delta \beta_{2}\delta \beta_{3}}{8\pi^{2}}\right]+R\sum_{i=1}^{n_{\mathrm{rot}}}\ln\left[\frac{\delta \varphi_{i}}{2\pi }\right]+R\sum_{j=1}^{n_{\mathrm{vibr}}}\ln\left[\frac{\delta \varphi_{j}}{n_{j}\Delta \alpha_{j}}\right].$$

Here, R is the gas constant, and the four terms of Equation (8) correspond to the four considered types of movements; see Figure 1.

The first term stands for the loss of translational entropy; $\delta x_{1}$, $\delta x_{2},$ $\delta x_{3}$ (measured in Å) are ranges of movements along three translational degrees of freedom in the solid phase (it is reasonable to set all of them equal to δx); here, a molecular movement is limited to a volume of $V_{\mathrm{crystal}}=\delta x_{1}\delta x_{2}\delta x_{3}$; ${\hat{V}}_{\mathrm{vapor}}=k_{B}T/P_{\mathrm{sat}.\mathrm{vapor}}$ is a volume per molecule in saturated vapor (with pressure $P_{\mathrm{vapor}}$ and temperature *T*), $k_{B}$ being the Boltzmann constant (the usage of ${\hat{V}}_{\mathrm{vapor}}\cdot e$ ⋅rather than simply ${\hat{V}}_{\mathrm{vapor}}$ following from standard statistical physics, as explained in Supporting Information to [26]).

The second term stands for the loss of entropy of rotations of the molecule as a whole; $\delta \beta_{1}$, $\delta \beta_{2}$, $\delta \beta_{3}$ are ranges of angles (in radians) of all three rotations of the molecule permitted by the solid phase; it is reasonable to set $\delta \beta_{k}=\delta x/A_{k}$, where $A_{1}$, $A_{2}$, $A_{3}$ (measured in Å) are three maximal radii of the molecule calculated from atomic coordinates (see Supporting Information to [26]).

The third term stands for the loss of entropy of rotations inside the molecule; $n_{\mathrm{rot}}$ is the number of “free” rotations around covalent bonds in the molecule in its free state (these rotations are free due to “low” barriers of the torsional potentials, whose heights γ < *k_B_T*/2 (see [26]); $\delta \varphi_{i}$ is the range of angles of rotation around bond *i* permitted by the solid phase; it is reasonable to set $\delta \varphi_{i}=\delta x/B_{i}$, where $B_{i}$ (measured in Å) is the maximal radius of the smallest group rotating around the covalent bond *i* (see Supporting Information to [26]).

The fourth term stands for the loss of vibrational entropy; $n_{\mathrm{vibr}}$ is the number of “soft vibrations” around covalent bonds. Vibrations around covalent bonds are rotations hindered, even in a free molecule, by high (γ ≥ *k_B_T*/2) barriers of torsional potentials. Some (“rigid”) vibrations (corresponding to potentials with γ > 35*k_B_T*; see [26]) are too small to be additionally hindered by crystals at room temperature; these “rigid” vibrations do not affect the sublimation entropy and are not taken into account here. Other vibrations with γ ≥ *k_B_T*/2 are “soft”: they are restricted by crystals and taken into account; $\delta \varphi_{i}=\delta x/B_{j}$ is the range permitted by the solid phase angles for vibrations around bond *j*; here, $B_{j}$ (similarly to $B_{i}$) is the maximal radius of the smallest group rotating around the corresponding covalent bond; $n_{j}$ is the number of energy minima for the torsion, in a free molecule, around bond *j*; as a rule, $n_{j}^{}$ = $n_{j}^{0}$, where $n_{j}^{0}$ is the multiplicity of the torsional potential, but $n_{j}^{}<n_{j}^{0}$ if some energy minima of the torsional potential are greatly elevated by other non-covalent interactions (such as the energy of *cis*-rotamer of the peptide bond); however, no such cases emerged for the molecules considered in this work; $\Delta \alpha_{j}$ is the range of angles of vibrations around bond *j* in the free state (see Supporting Information to [26]).

Now, the values of $\ln\left({\hat{V}}_{\mathrm{vapor}}\right)$ for all used compounds can be calculated from the above equation as:(9)$$\ln{\hat{V}}_{\mathrm{vapor}}=\frac{\Delta S_{\mathrm{subl}}}{R}-\ln\left[8\pi^{2}e\cdot A_{1}A_{2}A_{3}\right]-\sum_{i=1}^{n_{\mathrm{rot}}}\ln\left[2\pi B_{i}\right]-\sum_{j=1}^{n_{\mathrm{vibr}}}\ln\left[n_{j}\Delta \alpha_{j}B_{j}\right]+\left(6+n_{\mathrm{rot}}+n_{\mathrm{vibr}}\right)\bar{\ln\left[\delta x\right]}.$$

Having $-\Delta S_{\mathrm{subl}}=-\Delta H_{\mathrm{subl}}/T$ (see above), we can exclude $\Delta S_{\mathrm{subl}}$ (which is not measured directly), and obtain:(10)$$\ln{\hat{V}}_{\mathrm{vapor}}=\frac{\Delta H_{\mathrm{subl}}}{RT}-\ln\left[8\pi^{2}e\cdot A_{1}A_{2}A_{3}\right]-\sum_{i=1}^{n_{\mathrm{rot}}}\ln\left[2\pi B_{i}\right]-\sum_{j=1}^{n_{\mathrm{vibr}}}\ln\left[n_{j}\Delta \alpha_{j}B_{j}\right]+\left(6+n_{\mathrm{rot}}+n_{\mathrm{vibr}}\right)\bar{\ln\left[\delta x\right]}.$$

Here, as everywhere above, ${\hat{V}}_{\mathrm{vapor}}$ is expressed in Å^3^. However, the required concentration $\left[C_{\mathrm{vapor}}\right]$ is usually expressed not in 1/Å^3^, but in mol/L. Since 1 mole includes 6.02 × 10^23^ molecules, and 1 L includes 10^27^ Å^3^, 1 mol/L = 1/1660 Å^3^; therefore, log($\left[C_{\mathrm{vapor}}\;\left(\mathrm{mol}/L\right)\right]$) = log(1660) + log(1/${\hat{V}}_{\mathrm{vapor}}$) ≡ 3.220 + 0.4343 × ln(1/${\hat{V}}_{\mathrm{vapor}}$) (here, 3.220 = log(1660) and 0.4343 = log(*e*)). Thus, we obtain the predictions for crystal-to-vapor dissociation constant:(11)$$K_{D,\mathrm{vapor}}=\log\left[C_{\mathrm{vapor}}\left(\mathrm{mol}/L\right)\right]=3.220+\;0.4343\times \left\{-\frac{\Delta H_{\mathrm{subl}}}{RT}+\ln\left[8\pi^{2}eA_{1}A_{2}A_{3}\right]+\overset{n_{\mathrm{rot}}}{\underset{i=1}{\sum }}\ln\left[2\pi B_{i}\right]+\overset{n_{\mathrm{vibr}}}{\underset{j=1}{\sum }}\ln\left[n_{j}\Delta \alpha_{j}B_{j}\right]-\left(6+n_{\mathrm{rot}}+n_{\mathrm{vibr}}\right)\bar{\ln\left[\delta x\right]}\right\}.$$

Finally, for aqueous solutions, $K_{D,\mathrm{solution}}$ is equal to $K_{D,\mathrm{vapor}}\times k_{H,\mathrm{cc}}$ (Equation (7)).

### 2.3. Dataset

For this study, we took the same 28 compounds as in the previous works [26,27]. These 28 compounds were small organic molecules whose crystals melted at temperatures higher than +25 °C (namely, from +26.5 to 171 °C). This allowed working only with crystals that remained solid at the standard temperature of +25 °C.

A list of the selected 28 crystals of small organic molecules is given in Appendix A, with the experimental data on sublimation and dissolving. It should be noted that almost all of these organic molecules contained rigid cycles and, therefore, their crystals remained solid at +25 °C, while crystals of “flexible” molecules—such as alkanes—melted at temperatures much lower than +25 °C (see Appendix A to [26]).

## 3. Results

### 3.1. Crystal-Vapor Equilibrium

In our previous studies [25,26], we estimated (at the standard 25 °C, separately for each molecule from our set of 28) the range of molecular movements in crystals (δx) and the corresponding amplitudes; see Figure 2a. Then, we calculated the binding entropy (using the average value $\bar{\delta x}$ = 0.84 Å for all molecules and the above mentioned geometrical and energy parameters of the molecules), and then (using, in addition, the experimental sublimation enthalpies), we calculated the crystal-to-vapor dissociation constants as described in detail in our previous works [25,26], as well as in Appendix A.

However, the δx value, actually, has to be dependent on the strength of the binding of the molecule in the crystal, and the stronger the binding, the smaller the range of allowed movements should be. Accounting for the dependence of δx on the strength of binding (see the blue dashed line in Figure 2b) could, in principle, improve our estimate of the entropies and crystal-to-vapor dissociation constants. However, our attempts to obtain such an improvement led to such negligible results that we returned to the equal $\bar{\delta x}$ values for all molecules.

### 3.2. Vapor-Solution Equilibrium and Calculation of Dissociation Constants

For the second transition (vapor-to-solution), we only use the experimental Henry constant *k*_H,cc_ (see Ref. [29] and Equation (6)). Henry constants for each of the investigated 28 compounds were collected [16] from the literature and listed in Appendix A.

Now, given that we knew the value of *k*_H,cc_, and previously calculated [26] the crystal-to-vapor dissociation constant ($K_{D\;\mathrm{vapor}}=\left[C_{\mathrm{vapor}}\right]$), we can calculate the crystal-to-solvent dissociation constant as $K_{D\;\mathrm{solution}}=\left[C_{\mathrm{solution}}\right]=\left[C_{\mathrm{vapor}}\right]\times k_{H,\mathrm{cc}}$ (see Equation (6)). The result (in comparison with the experiment) is shown in Figure 3b. For comparison, Figure 3a shows, in the same format, the data for the crystal-to-vapor dissociation constant.

One can see that, in both cases, the points do not deviate far from the diagonal, which would be an ideal prediction. Thus, for most of the crystals, the predicted dissociation constants are closer to the experimental values than one order of magnitude: the average difference between the predicted and experimental values, i.e., $\langle |\log{\left[C_{\mathrm{solution}}\right]}_{\mathrm{predicted}}-\log{\left[C_{\mathrm{solution}}\right]}_{\mathrm{experimental}}|\rangle$, is 0.84, and the average $\langle |\log{\left[C_{\mathrm{vapor}}\right]}_{\mathrm{predicted}}-\log{\left[C_{\mathrm{vapor}}\right]}_{\mathrm{experimental}}|\rangle$ is 0.72. The correlation coefficient of the predicted values of dissociation constants with the experimental ones is 89% for the crystal–solvent dissociation constant, while it is 95% for the crystal–vapor dissociation constant.

When comparing the logarithms of experimentally measured ${\left[C_{\mathrm{solution}}\right]}_{\mathrm{experimental}}$ values with logarithms of products of the experimental values $\left[C_{\mathrm{vapor}}\right]$ and $k_{H,\mathrm{cc}}$, it appeared that they correlated at the level of 96% rather than 100% (see Appendix A), because they originated from different literature sources. When compare our calculated values of $\log{\left[C_{\mathrm{solution}}\right]}_{\mathrm{predicted}}$ with logarithms of products of the experimental values, $\left[C_{\mathrm{vapor}}\right]\times k_{H,\mathrm{cc}}$, the correlation coefficient is 93% (see Appendix A).

It is interesting to compare our results with those recently obtained [30] using molecular dynamics. Only two compounds were studied in [30] (because molecular dynamics simulations require an enormous amount of time); one of them was naphthalene, which was also present in our study. The authors of [30] investigated the dissolution of naphthalene not in water, but in two other solvents. Our method gave a twice larger value of the predicted concentration than the experimental one for the naphthalene dissolution in water. Molecular dynamics gave a 17% discrepancy with the experiment for the dissolution of naphthalene in toluene, and a 2.6 larger value (as compared to the experiment) for its dissolution in ethanol (see Table 1 in Ref. [30]). This example suggests that our simple method can provide results comparable to those given by molecular dynamics (but requires incomparably less time).

Thus, the developed method can successfully and rapidly predict dissociation constants for the transition of molecules not only from crystal to vapor, but also from crystal to solution.

## 4. Conclusions and Future Plans

In this work, we presented further development of our simple and fast approach for the estimation of molecular mobility in the solid state and the calculation of the entropy of binding of a vapor molecule to a crystal. We showed that the application of this approach to the sublimation of crystals, as well as to their dissolution, allows successfully predicting dissociation constants.

It is not out of place reminding of the limitations of the presented method (see [25,26]). This method should be, strictly speaking, valid for molecular crystals that are very stable at room temperature (and, thus, show small dissolution and vaporization), because it assumes that there is no interactions between the considered molecules both in vapor and in solution. It also assumes that there is no breakdown of molecules into pieces at sublimation and dissolution. Currently, some parameters used to calculate the dissociation constants (sublimation enthalpy and Henry constants) were taken from the experiment, and only the entropy was directly computed. In the future, we plan to replace these experimental data by calculations with the available force fields that take into account the interaction with the aqueous solvent [15,16] and, thus, to calculate the binding free energy and dissociation constants without using the experimental data. Finally, we plan to apply the developed approach to other, more complicated binding objects, including biomolecules (such as proteins with their ligands, aggregates of different types, etc.), as well as for more complicated phenomena such as phase transitions involving biomolecules [31].

## Figures and Tables

**Figure 1 biomolecules-12-00147-f001:**
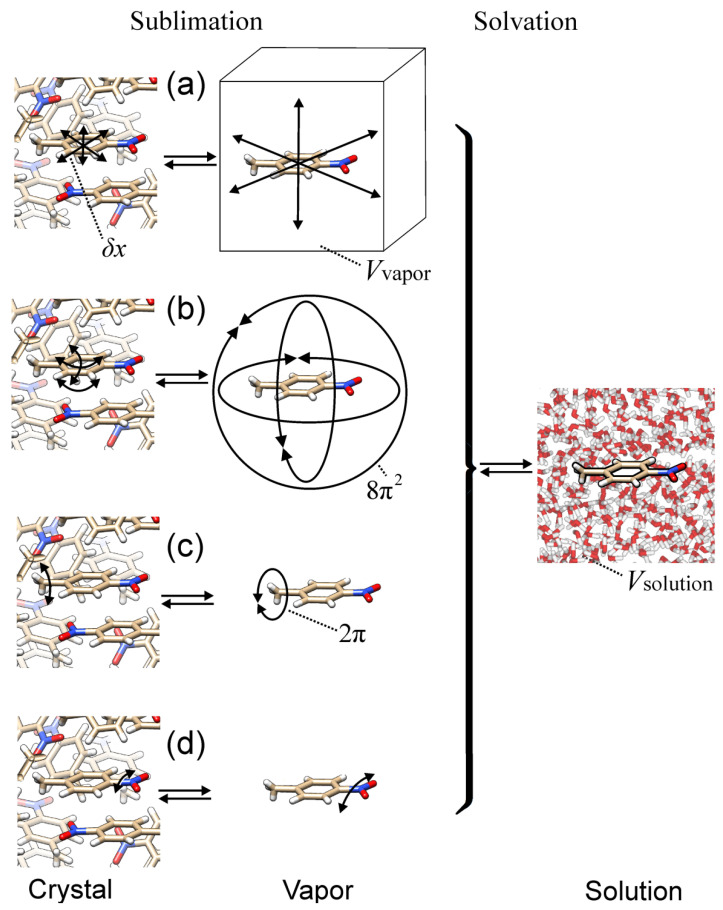
A scheme of the considered processes: reversible sublimation (from crystal (left column) to vapor (middle column)) and reversible solvation (from vapor (middle column) to aqueous solution (right column)). Four considered types of movements available in vapor (middle column) but hindered in crystal (left column): (**a**) translational movements of the considered molecule as a whole in vapor and the corresponding movements (vibrations) in a crystal; (**b**) rotational movements of the molecule as a whole in vapor and the corresponding vibrations in a crystal; (**c**) rotations around a covalent bond with a low potential barrier in vapor and the corresponding vibrations in a crystal; (**d**) vibrations around a covalent bond with a moderately high potential barrier in vapor and the corresponding vibrations in a crystal. δx is the range of movements of the considered molecule in crystal; $V_{\mathrm{vapor}}^{}$ and $V_{\mathrm{solution}}^{}$ are the volumes per crystal-forming molecule in vapor and in solution, respectively.

**Figure 2 biomolecules-12-00147-f002:**
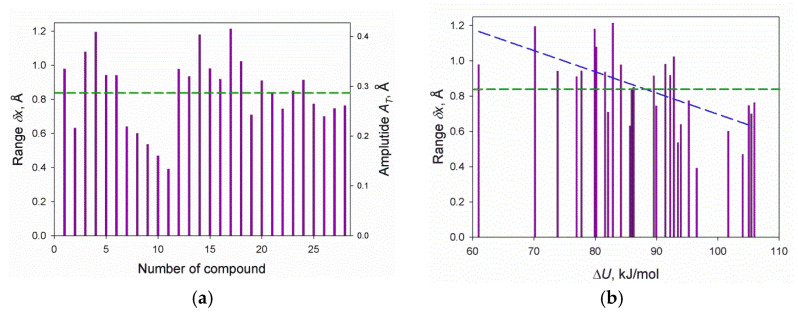
Calculated ranges δx and amplitudes *A_T_* = $\delta x/\sqrt{\pi e}$ (see Appendix A) for molecular movements in the 28 considered crystals (see Appendix A) arranged in the standard order, see Appendix A (**a**) and the change in ΔU, the potential energy of sublimation (**b**). The blue dashed line in (**b**) is the best-fit line. The green horizontal dashed line denotes the mean value, $\bar{\delta x}$ = 0.84 Å.

**Figure 3 biomolecules-12-00147-f003:**
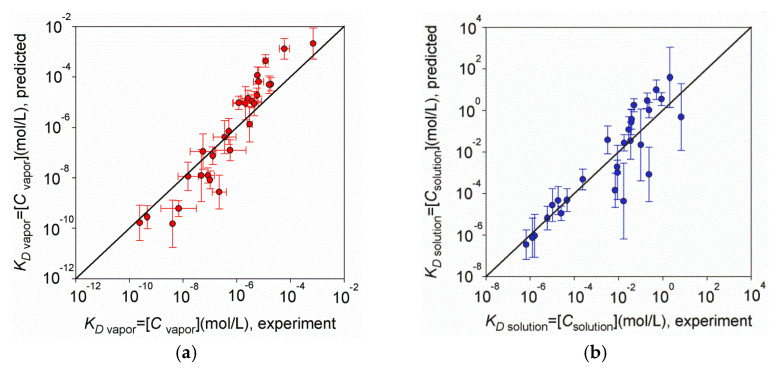
Calculated and experimental equilibrium concentrations for 28 substances in vapor (**a**) and in solution (**b**). Correlation coefficients between the calculated and experimental values were 95% and 89% in vapor and in solution, respectively. The diagonal solid line represents the “ideal prediction” (equal to the experimental data). The errors in experimental $\left[C_{\mathrm{vapor}}\right]$ values originated from errors in the experimental saturated vapor pressure *P*_vapor_ values (see Appendix A in Supporting Information to [26]). The errors in predicted $\left[C_{\mathrm{vapor}}\right]$ values originated from errors in experimental $\Delta H_{\mathrm{subl}}$ values (see ±*δ*$\langle \Delta H_{\mathrm{subl}}/RT\rangle$ in Appendix A) and in the average $\bar{\ln\delta x}$. The errors in predicted $\left[C_{\mathrm{solution}}\right]$ values originated from errors in experimental $\Delta H_{\mathrm{subl}}$ values, in the average $\bar{\ln\delta x}$, and in the experimental $k_{H,\mathrm{cc}}$ values (see ±*δ*$\langle \ln k_{H,\mathrm{cc}}\rangle$ in Appendix A). The errors in all experimental $\left[C_{\mathrm{solution}}\right]$ values were all within the size of symbols in Figure 3b.

## Data Availability

Not applicable.

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
