# Peer review of "Calculation of Crystal-Solution Dissociation Constants"

_biomolecules, 2022, doi:10.3390/biom12020147_

Round 1

Reviewer 1 Report

The topic reported by the authors is interesting and according to my opinion it probably deserves publication. However, there are two points that the authors should consider:

1- the results are not clearly presented; first of all the authors should separate the methods from the results;

2- it is not clear to me, how do they calculate the entropy.

Author Response

Thank you for careful reading of our manuscript and for your valuable advices! Please see below (as well as in the attachment) our answers to your comments and suggestions.

The topic reported by the authors is interesting and according to my opinion it probably deserves publication. However, there are two points that the authors should consider:

1- the results are not clearly presented; first of all the authors should separate the methods from the results;

 Response 1: Now, we have separated the methods and the results.

2- it is not clear to me, how do they calculate the entropy.

Response 2: Now, we have written (in the Methods section of the main text) how we calculate the entropy (previously, it was only done in the Supplementary Materials).

Kind regards,

Prof. Alexei V. Finkelstein,

[email protected]

Reviewer 2 Report

In this manuscript, the authors use a smart way to combine experimental Henry’s law constant with their crystal vapor equilibrium constant calculation to obtain the crystal solution equilibrium constant. For such a simple model, the results show promise. However, I would like the authors to clarify the following before accepting for publication.

In Figure 3, the authors compare their calculation results with experimental values. How were the error bars in the calculated results obtained? Also, even though experimental Henry’s constant is used, the correlation for the crystal solution equilibrium constant is worse than the crystal vapor equilibrium constant. Why is this?  Is this because the assumption of [Csolution] and [Cvapor] are small breaks down? If one uses the experimental crystal vapor equilibrium constant with the experimental Henry’s constant, how will that compare for the experimental crystal solution equilibrium constant?

Also, please quantify the value of concentration is “small” for [Csolution] and [Cvapor]? I guess this means that this approximation uses molecular crystals that are very stable at room temperature and show small dissolution or vaporization. Thereby, it will be good to clarify the limitations of the method that is used.

Lastly, I think it will be good to quantify the accuracy of a predicted entropy with their method and the usual MD simulations for a few important systems.

Author Response

Thank you for careful reading of our manuscript and for your important notes! Please see below (as well as in the attached file) our answers to your comments and suggestions.

In this manuscript, the authors use a smart way to combine experimental Henry’s law constant with their crystal vapor equilibrium constant calculation to obtain the crystal solution equilibrium constant. For such a simple model, the results show promise. However, I would like the authors to clarify the following before accepting for publication.

In Figure 3, the authors compare their calculation results with experimental values. How were the error bars in the calculated results obtained?

Response 1: In Figure 3, the error bars for the [Cvapor]predicted (Figure 3(a)) were calculated from errors in experimental ΔHsubl values (see ±d<ΔHsubl/RT> in Table S1) and in the average lndx value (while for the directly measured [Cvapor]experimental the experimental errors were marked), and in the case of solution (Figure 3(b)), the errors in [Csolution]predicted were calculated from the calculated above errors in [Cvapor]predicted and errors in experimental Henry’s law constants. Now, we add this information into the Figure 3 caption. Thank you very much for drawing our attention to the errors, because it appeared that the errors in [Csolution]experimental that were shown in Figure 3(b) were (our fault!) not for the directly measured [Csolution]experimental, but for the products [Cvapor]experimental*kH,cc, which are only approximately equal to [Csolution]experimental (see Response 2 below). Now, we have replaced them by the proper errors in [Csolution]experimental, which are in all cases within the size of the circles in Figure 3(b). Thank you again for your comment!

Also, even though experimental Henry’s constant is used, the correlation for the crystal solution equilibrium constant is worse than the crystal vapor equilibrium constant. Why is this?  Is this because the assumption of [Csolution] and [Cvapor] are small breaks down? If one uses the experimental crystal vapor equilibrium constant with the experimental Henry’s constant, how will that compare for the experimental crystal solution equilibrium constant?

Response 2: We have compared [Csolution]experimental and [Cvapor]experimental*kH,cc. They are not identical (because they have been taken from different sets of literature sources), and their correlation is 96% (see the new Figure S2); this difference answers to the question posed by the reviewer.

Also, please quantify the value of concentration is “small” for [Csolution] and [Cvapor]?

Response 3: We consider the concentration as “small” if it is below 1 mol/L. Now, it is mentioned in the text (after Eq.5).

I guess this means that this approximation uses molecular crystals that are very stable at room temperature and show small dissolution or vaporization. Thereby, it will be good to clarify the limitations of the method that is used.

Response 4: We have now added an explicit note on limitations of our method (in Conclusion).

Lastly, I think it will be good to quantify the accuracy of a predicted entropy with their method and the usual MD simulations for a few important systems.

 Response 5: We have added (in the last but one paragraph before Conclusions) a note on comparison of our results with the result of the recently-published paper on prediction of solubility of two organic molecules (one of which is present also in our database) by molecular dynamics simulation, which shows that our simple method gives the results not worse than molecular dynamics (and of course, it requires incomparably less time).

Kind regards,

Prof. Alexei V. Finkelstein,

[email protected]

Round 2

Reviewer 1 Report

The authors have properly revised the manuscript and addressed

all the comments.

Reviewer 2 Report

This is a revised manuscript combining experimental Henry’s law constant with a simple crystal vapor equilibrium constant model to obtain the crystal solution equilibrium constant. The authors have fully answered the questions raised by the referee, and the manuscript is acceptable for publication. One minor point is that in line 106, the chemical potential of crystal does not need the superscript of “-“